# Metal Concentration in Muscle and Digestive Gland of Common Octopus (*Octopus vulgaris*) from Two Coastal Site in Southern Tyrrhenian Sea (Italy)

**DOI:** 10.3390/molecules24132401

**Published:** 2019-06-29

**Authors:** Andrea Ariano, Raffaele Marrone, Rebecca Andreini, Giorgio Smaldone, Salvatore Velotto, Serena Montagnaro, Aniello Anastasio, Lorella Severino

**Affiliations:** 1Department of Veterinary Medicine and Animal Production, Division of Toxicology, University of Naples Federico II, Via Delpino 1, 80137 Naples, Italy; 2Department of Veterinary Medicine and Animal Production, Division of Food Inspection, University of Naples Federico II, Via Delpino 1, 80137 Naples, Italy; 3Department Veterinary Medical Sciences, University of Bologna, Viale Vespucci 2, 47042 Cesenatico (FC), Italy; 4Department of Agricultural Sciences, University of Naples, Federico II, via Università 100, 80055 Portici (NA), Italy; 5Department of Veterinary Medicine and Animal Production, Division of Infectious Deseases, University of Naples Federico II, Via Delpino 1, 80137 Naples, Italy

**Keywords:** lead, cadmium, mercury, cephalopods

## Abstract

*Octopus vulgaris* constitute an important part of most suitable marine resources for human consumption, however, they can represent a source in chemical contaminants intake such as heavy metals. In this scenario, the aim of the study was the evaluation of the concentration of lead (Pb), cadmium (Cd) and mercury (Hg) in the muscle and digestive gland of octopus caught from two different locations along Campania coast (Castellammare di Stabia and Napoli) and the estimation of their weekly human intake derived from the ingestion of octopus. Analysing 38 samples showed a higher concentration of Pb in the muscle of octopus in Castellammare di Stabia than in Napoli. No statistical differences were reported for Cd, Pb and Hg concentrations in the digestive gland of octopus between two sampling sites. Differences were observed between the two tissue types, with a higher level of Cd and Pb observed in the digestive gland compared with the muscle. Noteworthy, the consumption of muscle from Castellammare di Stabia could increase Pb intake in heavy consumers of local octopus. In conclusion, the present work determines that it is important to improve strategies to minimize environmental pollution sources in these areas.

## 1. Introduction

Regular dietary fishery products intake is recommended by nutritionists since they contain high concentrations of functional nutrients, including omega-3 fatty acids, useful in decreasing the risk of cardiovascular diseases [1]. Fishery products consumption in Italy has increased from 16 kg/year per person in 2016, to 25 kg/year in 2018, with good prospects of further growth [2]. Cephalopods constitute an important part of the marine resources most suitable for human consumption. Common octopus (*Octopus vulgaris*) is mainly consumed in Southern European countries such as Italy and Spain; consumption in Italy has a range from 1.5 to 5.1 kg per capita/year [3]. However, this species can represent a source for chemical contaminants intake. The levels of heavy metals in tissues of marine organisms is mainly influenced by biotic and abiotic factors [4]. *O. vulgaris* is a benthic species, living in direct contact with the seabed, which constitutes a possible pathway for trace element accumulation, and can therefore represents a source of human exposure to toxic elements [5,6].

In the Campania region (Italy) several areas exist where the pollution of soil, marine water, and groundwater is extremely severe and represents a serious hazard to public health. Moreover, these sites are located near highly urbanized and populated areas and usually represented by ex-industrial areas or lands nearby illegal waste dumps [7].

European Regulation 1881/06 [8] established maximum levels of contaminants in foodstuffs, fixing specific limit for heavy metals in Cephalopods (without viscera). The Joint FAO/WHO Expert Committee on Food Additives [9] revised its risk assessment on heavy metals in fish and adopted a PTWI of 4µg/kg b.w. week for mercury, 7 µg/kg b.w. for cadmium and 25 µg/kg b.w. for lead. Therefore, the objectives of the present study were: firstly to evaluate lead, cadmium and mercury levels in the muscle of common octopus (*Octopus vulgaris*) collected at two different sites along the Southern Tyrrhenian Sea coast (Italy) and estimate the weekly human intake (WHI) of heavy metals deriving from the ingestion of octopus compared with the PTWI; secondly to investigate the geographical variation of the same metals in muscle and digestive gland.

## 2. Results and Discussion

Mean concentrations of heavy metals in samples of the muscle and digestive gland of Octopus vulgaris are summarized in Figure 1. Results showed significantly higher concentration of Pb (*p* < 0.001) in the muscle of *O. vulgaris* in Castellammare di Stabia (mean value 0.537 µg g^−1^) versus Napoli (mean value 0.046 µg g^−1^). Levels of Cd and Hg were very low in all samples of octopus muscle. No statistical differences were reported for Cd and Hg concentrations in the muscle of octopus between both sampling sites. In the digestive gland Cd was found at a mean concentration of 2.643 and 2.969 µg g^−1^, Pb was found at a mean concentration of 1.511 and 0.763 µg g^−1^ and Hg of 0.040 and 0.101 µg g^−1^, in Castellammare di Stabia and Napoli, respectively. No statistical differences were reported for Cd, Pb and Hg concentrations in digestive gland of octopus between Castellammare di Stabia and Napoli (Figure 1). 

Metal concentrations in tissues of *O. vulgaris* captured along the Campania coast were compared to the values reported for *O. vulgaris* in other coastal waters.

In the present study, Hg concentrations in muscle of *O. vulgaris* were lower than levels found in samples from the Portuguese coast (0.49 µg g^−1^) [10]. Also, other works reported higher levels of Hg in the muscle (0.13–0.76 µg g^−1^) and digestive gland (0.36–7.4 µg g^−1^) [11,12].

Lead concentration in the in muscle was higher than levels reported in samples from the Portuguese coast, ranging from 0.04 to 0.09 µg g^−1^ [10]. However, other studies showed higher levels of Pb (range value 1.5–7.2 µg g^−1^) in the digestive gland of *O. vulgaris* than those reported in the current study [13,14].

### 2.1. Metal Concentration versus Sampling Sites and Tissue Type

Data analysis using multivariate tests allowed an estimation of how tissue type and sampling sites influence heavy metals concentration (Table 1). 

The tissue type had a highly significant (*p* < 0.001) influence on the accumulation of Cd and Pb, leading to higher accumulation in the digestive gland (mean values of 2.81 and 1.137 µg g^−1^, respectively) than in muscle (Figure 2A). Mercury was accumulated at same concentration in both organs. Significant site-specific differences were detected for Pb (*p* < 0.01) only. Concentration of lead were significatively higher in octopus collected in Castellammare di Stabia (1.024 µg g^−1^) versus that in Napoli (0.405 µg g^−1^) (Figure 2B).

All digestive gland samples of octopus showed the highest Cd and Pb concentration, confirming the primary role of this district in the bioaccumulation and detoxification processes of Cd and Pb and confirming the presence of these metals at both sampling areas [6,15]. Also, a similar distribution of Hg in the two different tissue has already been described in other studies [15,16].

Cadmium concentrations in muscle and digestive gland in the present study were comparable with those previously reported by other authors [10,17].

### 2.2. Metal Concentration versus Biological Parameters

The analyzed individuals varied in size and weight ranges, including males and females. The multiple regression analyses indicate that there was no correlation between weight, gender and concentration of Pb, Cd and Hg (*p* > 0.05). The lack of relations between heavy metals concentration in muscle and digestive gland of *O. vulgaris* and weight suggest that, within the range of weight of the studied specimens these parameters had minor effect on metal accumulation. These observations were already described in other studies on *O. vulgaris* captured along Portuguese coast [13]. The absence of relation between Pb, Cd and Hg levels and gender agrees with other studies on cephalopods [13,18].

### 2.3. Concern for Public Health

Levels of Cd and Hg in present study were low in all samples of muscle and were below the legal limit for human consumption. Anyway, the average concentrations of Pb in samples of muscle from Castellammare di Stabia were above the maximum concentration level of 0.3 µg g^−1^, leading to the exclusion of this product for human consumption [8].

To establish possible human health implications related to consumption of octopus, the Pb estimate weekly intakes (EWI) were subsequently compared with the provisional tolerable weekly intake (PTWI) of 25 µg/kg of body weight [9]. Estimating a weekly consumption of 100 g of *O. vulgaris* muscle, EWI values were found to be 53.7 µg/week. This value accounted for 3.06% of the tolerable weekly intake set by EFSA. Considering the level of Pb, the consumption of octopus muscle from Castellammare di Stabia may increase Pb intake, but it would not contribute significantly to the PTWI. In contrast, it may contribute greatly to high EWI values in heavy consumer of octopus, when all other main contributors to dietary Pb intake and professional exposure were included in the exposure assessment.

## 3. Materials and Methods

### 3.1. Sampling

Thirty-eight samples of common octopus (*Octopus vulgaris*) were fished directly from two different locations along Campania coast (Italy) in autumn of 2016 (Figure 3). 

Once captured, the octopus were weighed, their total length were measured and then they were immediately sealed in individual polyethylene bags, frozen at −20 °C and kept at the same temperature until dissection. Sex was also determined for each individual (Table 2).

In the laboratory, the digestive gland of each organism was totally removed under partially defrost conditions without rupture of the outer membrane. Subsequently, the digestive gland was treated separately from the remaining tissues and an interior portion was sampled for metal analysis. Arms and mantle were dissected including the skin. Each tissue was subsequently homogenized by means of a laboratory mixer and stored at −20 °C until further analyses.

### 3.2. Chemical and Instrumental Analysis

Glassware and laboratory equipments were decontaminated before use with diluted ultrapure 65% HNO_3_ (Romil UpA, Cambridge, UK) and rinsed with Milli Q water (Millipore, Bedford, MA, USA). Aliquots of each sample (0.50 ± 0.02 g) were digested in 5 mL of ultrapure 65% HNO_3_ and 2 mL of 30% H_2_O_2_ (Romil UpA, Cambridge, UK) in a microwave digestion system (Milestone, Bergamo, Italy). The final volume was obtained by adding Milli-Q water. Metal concentrations in the digested samples were determined with an atomic absorption spectrometer (Aanalyst 600, Perkin-Elmer, Madrid, Spain) equipped with a graphite furnace and a L’vov platform for Pb and Cd. A flow injection analysis hydride system (FIAS 100, Perkin-Elmer) was used for the determination of the total Hg. The equipment was calibrated with standard solutions (Perkin-Elmer), resulting in a calibration curve with three concentrations for Pb and Hg and four concentrations for Cd. 

Recovery of the metals was determined by adding known amounts to metal-free samples, which were then subjected to the same digestion procedure. The resulting solutions were analyzed for metal concentrations. Recovery of metals from spiked samples ranged from 85 to 120%. Concentrations of each heavy metal were expressed as milligrams per kilogram wet weight [19].

### 3.3. Quality Assurance

Quality was monitored through analysis of procedural blanks, duplicate samples, and standard solutions. Standard solutions of analytes were prepared from certified stock solutions of Cd, Pb, and Hg with a relative matrix modifier (atomic spectroscopy standard, Perkin Elmer). Concentrations for each set of samples were determined in the medium range of the calibration curve.

The performance of the method was assessed through participation in interlaboratory studies organized by FAPAS (Food Analysis Performance Assessment Scheme, Sand Hutton, UK). The FAPAS studies were conducted with fish tissue. The limit of detection (LOD) and the limit of quantification (LOQ) were calculated by determining the standard deviation of 10 independent blanks spiked at 1, 2, 4 and 8 µg g^−1^ for Cd and 25, 50 and 100 µg g^−1^ for both Pb and Hg, with an external standardization curve [20,21,22].

### 3.4. Statistical Analysis

All metal concentrations were expressed in wet weight as mean ± SEM (standard error from mean) [16]. Factorial analysis of variance was used to test statistical significance of the influence of the sampling site (Napoli versus Castellammare di Stabia), the target tissue for metal accumulation (muscle versus digestive gland) on the concentration of heavy metals. Moreover, ANOVA and Mann-Whitney test was used to detect differences between metal concentration in muscle and digestive gland and sampling area. Statistical significance between concentration of metals and the variables (total weight and gender) were analysed using multiple regression. 

The normal distribution of data was confirmed by the One-Sample Kolmogorov-Smirnov Test. Statistical analyses were performed using MedCalc for Windows, version 18.11.3 (MedCalc Software, Ostend, Belgium). The result, of *p* < 0.05, was considered significant.

## 4. Conclusions

The present study provided data on heavy metals concentrations in Octopus vulgaris from the Southern Tyrrhenian Sea (Italy). The concentration of Cd and Pb found in tissues of octopus sampled at Castellammare di Stabia and Napoli witness for their presence in the environment. Mercury occurred only at trace concentrations at both sampling area. The results also indicate that there was no correlation between weight, gender and concentration of heavy metals. Instead, large variations in Cd and Pb concentration existed across muscle and digestive gland of this species. 

Considering Hg and Cd intake, the consumption of octopus muscle from both sampling sites does not contribute significantly to the PTWI of 4 µg/kg b.w. week and 7 µg/kg b.w. week, respectively. In contrast, the consumption of muscle from Castellammare di Stabia could increase Pb intake in the heavy consumer of local octopus, but it would not contribute significantly to the PTWI of 25 µg/kg b.w. week. 

The capability of digestive gland to accumulate higher levels of Cd than muscle, as reported by Roldán-Wong et al. [23] could provide a new tool for the monitoring of the geographical distribution of this metal, even when present at negligible levels in the edible part. The presence of Pb at a higher concentration at Castellammare di Stabia in both muscle and digestive gland was probably due to a major anthropogenic pressure in this site.

Monitoring studies on heavy metals in a greater number of samples of Octopus vulgaris and other species of the marine food chain, as suggested by Sangiuliano et al. [24], will provide more detailed information of the human exposure to metals in these areas. Although monitoring of levels of chemical contaminants in marine food resources is fundamental for human health and ecological approach, it is needful to improve strategies to minimize environmental pollution sources in these areas.

## Figures and Tables

**Figure 1 molecules-24-02401-f001:**
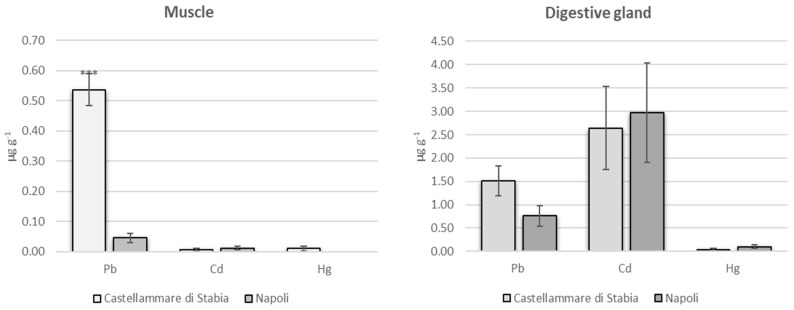
Mean concentration (µg g^−1^ wet weigth) ± SEM of Pb, Cd and Hg in *O. vulgaris* muscle and digestive gland from Napoli and Castellammare di Stabia. Probability levels for significant differences from sampling sites: *p* < 0.001(***).

**Figure 2 molecules-24-02401-f002:**
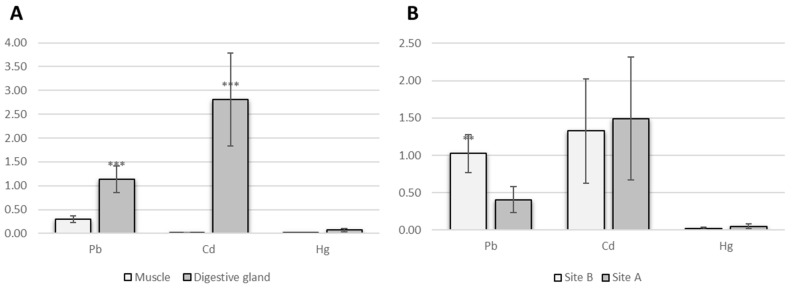
Concentration of Pb, Cd and Hg in *O. vulgaris* depending on (**A**) organ type: muscle versus digestive gland; (**B**) sampling sites: Napoli versus Castellammare di Stabia. Vertical bars represent average concentration (µg g^−1^ wet weigth) ± SEM. Probability levels for significant differences: *p* < 0.001(***); *p* < 0.01 (**).

**Figure 3 molecules-24-02401-f003:**
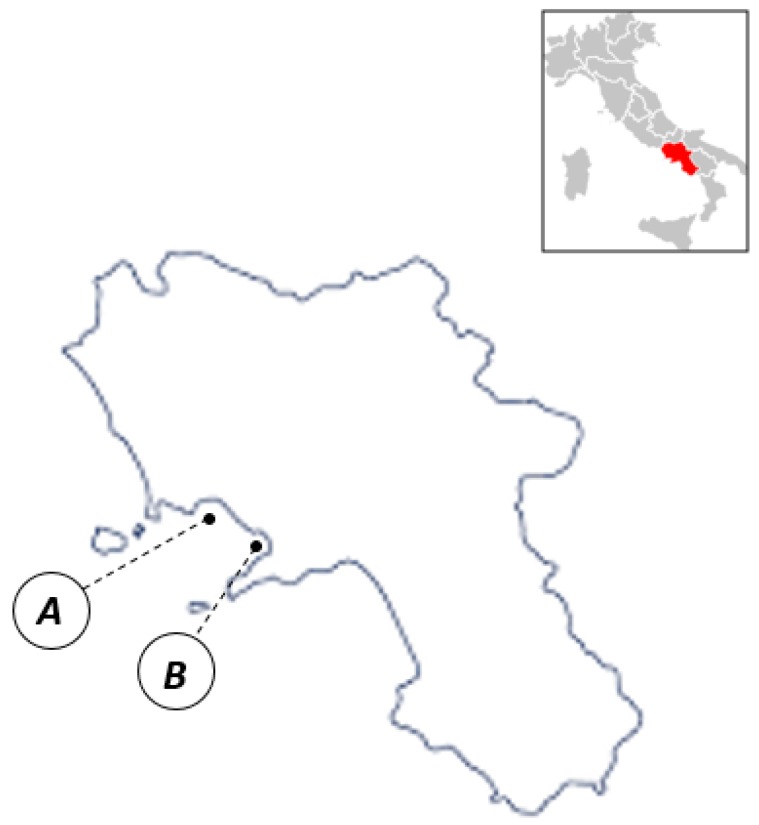
Map showing locations of the sampling sites: Napoli (site A) and Castellammare di Stabia (site B).

**Table 1 molecules-24-02401-t001:** Factorial analysis of variance (ANOVA) testing the effect from the collection site (Napoli versus Castellammare di Stabia), the accumulation organ type (muscle versus digestive gland) on the concentration of heavy metals (Pb, Cd and Hg) in Octopus vulgaris. df = degree of freedom. Probability levels for significant effects: *p* < 0.001 (***); *p* < 0.01 (**). MS = mean squares; F = F-ratio.

Source of Variation	Dependent Variable	df	Mean Square	F
Site	Pb	1	7.292	10.08 **
	Cd	1	0.521	0.06
	Hg	1	0.012	1.15
Organ	Pb	1	13.583	18.77 ***
	Cd	1	148.532	16.08 ***
	Hg	1	0.081	7.68

**Table 2 molecules-24-02401-t002:** Number of individuals (n), weight (g), size (mm) and sex of *O. vulgaris* captured along the Campania coast.

Sites	Geographic Coordinates	*n*	Weight Range (g)	Total Lenght Range (cm)	Sex
*Napoli*	40° 49′ 39″ N; 14°14′ 47″ E	19	845 ± 143	71.7 ± 18.3	15 ♀ 4 ♂
*Castellammare di Stabia*	40° 41′ 46″ N; 14°27′ 53″ E	19	740 ± 229	67.6 ± 21.5	8 ♀ 11 ♂

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
