# Peer review of "Metal Concentration in Muscle and Digestive Gland of Common Octopus (Octopus vulgaris) from Two Coastal Site in Southern Tyrrhenian Sea (Italy)"

_molecules, 2019, doi:10.3390/molecules24132401_

Round 1

Reviewer 1 Report

The authors describes the results of heavy metal research in octopus vulgaris from two different zones in the mediterranean coast (Italy). The results are interesting because deal with an important topic. There are several researchs with contradictory results, including interesting data regarding digestive gland and comparison between contamination in the different localizations and parts of the octopus. I think that the manuscript can be improved with some items indicated below.

some papers could be cited to improve the discussion. I think it is interesting to discuss data with results of

D. Sangiuliano, C. Rubio, A. J. Gutiérrez, D. González-Weller, C. Revert, A. Hardisson, E. Zanardi, and S. Paz (2017) Metal Concentrations in Samples of Frozen Cephalopods (Cuttlefish, Octopus, Squid, and Shortfin Squid): An Evaluation of Dietary Intake. Journal of Food Protection: November 2017, Vol. 80, No. 11, pp. 1867-1871. 

and

Roldán-Wong, N.T., Kidd, K.A., Ceballos-Vázquez, B.P. et al. Bull Environ Contam Toxicol (2018) 101: 796.

https://doi.org/10.1007/s00128-018-2447-9

The authors also must indicate how were sampled the octopus. Were fished? directly or take from local market?

Author Response

Molecules-529188

Dear Editor,

Enclosed the manuscript by Andrea Ariano, Raffaele Marrone, Rebecca Andreini, Giorgio Smaldone, Salvatore Velotto, Serena Montagnaro, Aniello Anastasio  and Lorella Severino entitled  “Metal concentration in muscle and digestive gland of common octopus (Octopus vulgaris) from two coastal site in Southern Tyrrhenian Sea (Italy)”.

I would like to thank the Reviewers for examining our paper in depth, for the interest shown, and for the comments that leaded us to improve the manuscript. The English language has been carefully reviewed as recommended.

In the body of the text the changes are highlighted in yellow.

Reviewer 1:

1. Some papers could be cited to improve the discussion. I think it is interesting to discuss data with results of

·         Sangiuliano, D.;  C. Rubio, C.; Gutiérrez, A. J.; González-Weller, D.; Revert, C.; Hardisson, A.; Zanardi, E.; Paz, S. Metal Concentrations in Samples of Frozen Cephalopods (Cuttlefish, Octopus, Squid, and Shortfin Squid): An Evaluation of Dietary Intake. J Food Protect 2017, 80, 11, 1867-1871.

·         Roldán-Wong, N.T.; Kidd, K.A.; Ceballos-Vázquez, B.P.; Arellano-Martínez, M. Is There a Risk to Humans from Consuming Octopus Species from Sites with High Environmental Levels of Metals? Bull Environ Contam Toxicol 2018, 101, 6, 796 - 802.

2. The authors also must indicate how were sampled the octopus. Were fished? directly or take from local market?

Authors

Thanks for your valuable comments and suggestions which have led to significant improvement on the presentation and quality of our manuscript.

1. Suggested references were added (see line 215 and 220)

2. The samples were directly fished (see Materials and Methods line 144)

Best wishes,

Dr  Giorgio Smaldone

Author Response

Molecules-529188

Dear Editor,

Enclosed the manuscript by Andrea Ariano, Raffaele Marrone, Rebecca Andreini, Giorgio Smaldone, Salvatore Velotto, Serena Montagnaro, Aniello Anastasio  and Lorella Severino entitled  “Metal concentration in muscle and digestive gland of common octopus (Octopus vulgaris) from two coastal site in Southern Tyrrhenian Sea (Italy)”.

I would like to thank the Reviewers for examining our paper in depth, for the interest shown, and for the comments that leaded us to improve the manuscript. The English language has been carefully reviewed as recommended.

In the body of the text the changes are highlighted in yellow.

Reviewer 2

Specific comments

1.       Title requires revision and monotonously used “from” should be reduced. The word “southern” must also be corrected with the capitalized “S”

2.       The authors are advised to be consistent with the use of the word “bw or b.w.” on page 7, line 58.

3.       Page 7, line 61, use lowercase “w” for weekly.

4.       The authors are advised to use the following reference for determination of toxic metals such as Pb, Cd etc. and their possible hazardous effects when consumed by the human beings. Additional reference for citation include the simultaneous determination of metals and anions in the high salinity environments

·         Lebea N. Nthunya, Monaheng L. Masheane, Soraya P. Malinga, Edward E. Nxumalo, Bhekie B. Mamba, and Sabelo D. Mhlanga. Determination of toxic metals in drinking water sources in the Chief Albert Luthuli Local Municipality in Mpumalanga, South Africa. Physics and Chemistry of the Earth, Parts A/B/C. 2017; 100, 94 – 100.

·         Lebea N. Nthunya, Sebabatso Maifadi, Bhekie B. Mamba, Arne R. Verliefde and Sabelo D. Mhlanga. Spectroscopic Determination of Water Salinity in Brackish Surface Water in Nandoni Dam, at Vhembe District, Limpopo Province, South Africa. Water. 2018; 10(990), 1-13.

5.       The authors are encouraged to provide the geographic coordinates of the sampling sites.

6.       Ensure that there is spacing between the punctuation marks and the first word of the next sentence throughout the entire manuscript.

7.       Revise the sentence on line 84

8.       “than” or “then” on line 86

9.       The sentence on lines 128-129 must be revised

10.    ”In contrast, may reach high EWI values in the heavy consumer of octopus, when the other main contributors to dietary Pb intake and professional exposure were included in the exposure assessment.” This sentence is confusing. Line 138

11.    Line 208: Revise this sentence “Regarding Hg and Cd intake, the consumption of octopus muscle from both sampling sites not contribute significantly to the PTWI of 4μg/kg b.w. week and 7 μg/kg b.w. week, respectively.”

Thanks for your valuable comments and suggestions which have led to significant improvement on the presentation and quality of our manuscript.

1.       As suggested by revisor, title was changed (see line 2-4)

2.       As suggested by revisor, bw was changed in b.w.  in all body text

3.       As suggested by revisor, lowercase “w” for weekly was used (see line 61)

4.       As suggested by revisor, references were added in the Materials and Methods section (see line 190)

5.       As suggested by revisor, the geographic coordinates of the sampling sites were added (see line tab. n. 2, line 157)

6.       As suggested by revisor, spacing between the punctuation marks and the first word of the next sentence throughout the entire manuscript was checked.

7.       As suggested by revisor, sentence on line 84 was revised (see line 84-84)

8.       As suggested by revisor, then was changed in than (see line 86)

9.       As suggested by revisor, sentence on lines 129-130 was revised

10.    As suggested by revisor, sentence on lines 139-141 was revised

11.    As suggested by revisor, sentence on lines 209-210 was revised

Best wishes,

Dr  Giorgio Smaldone
